# ERGODIC INFERENCE:
# ACCELERATE CONVERGENCE BY OPTIMISATION

## ABSTRACT

Statistical inference methods are fundamentally important in machine learning. Most state-of-the-art inference algorithms are variants of Markov chain Monte Carlo (MCMC) or variational inference (VI). However, both methods struggle with limitations in practice: MCMC methods can be computationally demanding; VI methods may have large bias. In this work, we aim to improve upon MCMC and VI by a novel hybrid method based on the idea of reducing simulation bias of finite-length MCMC chains using gradient-based optimisation. The proposed method can generate low-biased samples by increasing the length of MCMC simulation and optimising the MCMC hyper-parameters, which offers attractive balance between approximation bias and computational efficiency. We show that our method produces promising results on popular benchmarks when compared to recent hybrid methods of MCMC and VI.

## 1 INTRODUCTION

Statistical inference methods in machine learning are dominated by two approaches: simulation and optimisation. Markov chain Monte Carlo (MCMC) is a well-known simulation-based method, which promises asymptotically unbiased samples from arbitrary distributions at the cost of expensive Markov simulations. Variational inference (VI) is a well-known method using optimisation, which fits a parametric approximation to the target distribution. VI is biased but offers a computationally efficient generation of approximate samples.

There is a recent trend of hybrid methods of MCMC and VI to achieve a better balance between computational efficiency and bias. Hybrid methods often use MCMC or VI as an algorithmic component of the other. In particular, Salimans et al. (2015) proposed a promising modified VI method that reduces approximation bias by using MCMC transition kernels. Another technique reduces the computational complexity of MCMC by initialising the Markov simulation from a pre-trained variational approximation (Hoffman, 2017; Han et al., 2017). Levy et al. (2018) proposed to improve MCMC using flexible non-linear transformations given by neural networks and gradient-based auto-tuning strategies.

In this work, we propose a novel hybrid method, called ergodic inference (EI). EI improves over both MCMC and VI by tuning the hyper-parameters of a flexible finite-step MCMC chain so that its last state sampling distribution converges fast to a target distribution. EI optimises a tractable objective function which only requires to evaluate the logarithm of the unnormalized target density. Furthermore, unlike in traditional MCMC methods, the samples generated by EI from the last state of the MCMC chain are independent and have no correlations. EI offers an appealing option to balance computational complexity vs. bias on popular benchmarks in machine learning. Compared with previous hybrid methods, EI has following advantages:

- EI's hyperparameter tuning produces sampling distributions with lower approximation bias.
- The bias is guaranteed to decrease as the length of the MCMC chain increases.
- By stopping gradient computations, EI has less computational cost than related baselines.

We also state some disadvantages of our method:

- The initial state distribution in EI's MCMC chain has to have higher entropy than the target.

- The computational complexity per simulated sample of EI is in general higher than in VI.

## 2 BACKGROUND

### 2.1 MONTE CARLO STATISTICAL INFERENCE

Monte Carlo (MC) statistical inference approximates expectations under a given distribution using simulated samples. Given a target distribution $\pi$, MC estimations of an expectation $\mathbb{E}_\pi[f(\mathbf{x})]$ are defined as empirical average of the evaluation of $f$ on samples from $\pi$. To generate samples from $\pi$, we assume that the unnormalized density function $\pi^*(\mathbf{x})$ can be easily computed. In a Bayesian setting we typically work with $\pi^*(\mathbf{x}|\mathbf{y})$ given by the product of the prior $p(\mathbf{x})$ and the likelihood $p(\mathbf{y}|\mathbf{x})$, where $\mathbf{y}$ denotes observed variables and $\mathbf{x}$ denotes the model parameters specifying $p(\mathbf{y}|\mathbf{x})$.

### 2.2 MARKOV CHAIN MONTE CARLO

Markov chain Monte Carlo (MCMC) casts inference as simulation of ergodic Markov chains that converge to the target $\pi$. The MCMC kernel $M(\mathbf{x}'|\mathbf{x})$ is characterised by the detailed balance (DB) property: $\pi(\mathbf{x})M(\mathbf{x}'|\mathbf{x}) = \pi(\mathbf{x}')M(\mathbf{x}|\mathbf{x}')$. Given an unnormalised target density $\pi^*$, an MCMC kernel can be constructed in three steps: first, sample an auxiliary random variable $\mathbf{r}$ from an auxiliary distribution $q_{\phi_1}$ with parameters $\phi_1$; second, create a new candidate sample as $(\mathbf{x}', \mathbf{r}') = f_{\phi_2}(\mathbf{x}_{t-1}, \mathbf{r})$, where $f_{\phi_2}$ is a deterministic function with parameters $\phi_2$; finally, accept the proposal as $\mathbf{x}_t = \mathbf{x}'$ with probability $p_{\text{MH}} = \min\{0, \pi^*(\mathbf{x}')q_{\phi_1}(\mathbf{r}')/[\pi^*(\mathbf{x}_{t-1})q_{\phi_1}(\mathbf{r})]\}$, otherwise duplicate the previous sample as $\mathbf{x}_t = \mathbf{x}_{t-1}$. The last step is well known in the literature as the Metropolis-Hastings (M-H) correction step (Robert & Casella, 2005) and it results in MCMC kernels that satisfy the DB condition. In the following, we denote the joint MCMC parameters $(\phi_1, \phi_2)$ by $\phi$. If $f_{\phi_2}$ does not preserve volume, then it requires a Jacobian correction factor in the ratio in $p_{MH}$.

Hamiltonian Monte Carlo (HMC) is a successful MCMC method which has drawn great attention. A few recent works based on this method are Salimans et al. (2015); Hoffman (2017); Levy et al. (2018). In HMC, the auxiliary distribution $q_{\phi_1}$ is often chosen to be Gaussian with zero-mean and a constant diagonal covariance matrix specified by $\phi_1$. The most common $f_{\phi_2}$ in HMC is a numeric integrator called the leapfrog algorithm, which simulates Hamiltonian dynamics defined by $\log \pi^*$ (Neal, 2010). The leapfrog integrator requires the gradient of $\log \pi^*$ and a step size parameter given by $\phi_2$.

Given any initial state $\mathbf{x}_0$, MCMC can generate asymptotically unbiased samples $\mathbf{x}_{1:n}$. For this, MCMC iteratively simulates the next sample $\mathbf{x}_t$ through the application of the MCMC transition kernel to the previous sample $\mathbf{x}_{t-1}$. It is well known in the literature that MCMC is computationally demanding (MacKay, 2002; Bishop, 2006). In particular, it is often necessary to run sufficiently long burn-in MCMC simulations to reduce simulation bias. Another drawback of MCMC is sample correlation, which increases the variance of the MC estimator (Neal, 2010). To avoid strong sample correlation, the common practice in MCMC to tune hyper-parameters manually using sample quality metrics like effective sample size, (Hoffman, 2017; Robert & Casella, 2005), which has been developed into automated gradient-based tuning strategies in recent work (Levy et al., 2018).

### 2.3 MC ESTIMATION USING VARIATIONAL INFERENCE

Variational inference (VI) is a popular alternative to MCMC for generating approximate samples from $\pi$. Unlike MCMC reducing sample bias by long burn-in simulation, VI casts the sample bias reduction as an optimisation problem, where a parametric approximate sampling distribution $P$ is fit to the target $\pi$. In particular, VI optimises the evidence lower bound (ELBO) given by

$$L_{\text{ELBO}}(P \parallel \pi^*) = \mathbb{E}_p[\log \pi^*(\mathbf{x})] + H(P), \tag{1}$$

where $H(P) = -\mathbb{E}_p[\log p(\mathbf{x})]$, also known as the entropy, must be tractable to compute. $L_{\text{ELBO}}(P_T \parallel \pi^*)$ is a lower bound on the log normalising constant $\log Z = \log \int \pi^*(\mathbf{x})\, d\mathbf{x}$. This bound is tight when $P = \pi$. therefore, the approximation bias in VI can be defined as the gap between $L_{\text{ELBO}}(P \parallel \pi^*)$ and $\log Z$, that is,

$$\Delta_{\text{bias}}(P) = \log Z - L_{\text{ELBO}}(P \parallel \pi) = D_{\text{KL}}(P \parallel \pi) \geq 0, \tag{2}$$

where $D_{\text{KL}}(P \parallel \pi)$ denotes the Kullback-Leibler (KL) divergence.

Variational approximations often belong to simple parametric families like the multivariate Gaussian distribution with diagonal covariance matrix. This results in computationally efficient algorithms for bias reduction and sample generation, but may also produce highly biased samples in cases of over-simplified approximation that ignores correlation. Designing variational approximation to achieve low bias under the constraint of tractable entropy and efficient sampling procedures is possible using flexible distributions parameterised by neural networks (NNs) (Rezende & Mohamed, 2015; Kingma et al., 2016). However, how to design such NNs for VI is still a research challenge.

### 2.4 Hybrid Methods and Variants of MCMC and VI

The balance between computational efficiency and bias is a challenge at the heart of all inference methods. MCMC represents a family of simulation-based methods that guarantee low-bias samples at cost of expensive simulations; VI represents a family of optimisation-based methods that generate high-bias samples at a low computational cost.

Many recent works seek a better balance between efficiency and bias by combining MCMC and VI. Salimans et al. (2015) proposed to reduce variational bias by optimising an ELBO specified in terms of the tractable joint density of short MCMC chains. The idea seems initially promising, but the proposed ELBO becomes looser and looser as the chain grows longer. Caterini et al. (2018) construct an alternative ELBO for HMC that still has problems since the auxiliary momentum variables are sampled only once at the beginning of the chain, which reduces the empirical performance of HMC. Inspired by contrastive divergence, Ruiz & Titsias (2019) proposed a novel variational objective function to optimise variational parameters by adding additional term that minimise the KL between a MCMC distribution and variational approximation to reduce variational bias.

Hoffman (2017) and Han et al. (2017) proposed to replace expensive burn-in simulations in MCMC with samples from pre-trained variational approximations. This approach is effective at finding good initial proposal distributions. However, it does not offer a solution for tuning HMC parameters (Hoffman, 2017), which are critical for good empirical performance.

Another line of research has focused on improving inference using flexible distributions, which are transformed from simple parametric distributions by non-linear non-volume preserving (NVP) functions. Levy et al. (2018) proposed to tune NVP parameterised MCMC w.r.t. a variant of the expected squared jumped distance (ESJD) loss proposed by Pasarica & Gelman (2010). Song et al. (2017) proposed a similar auto-tuning for NVP parameterised MCMC using an adversarial loss.

## 3 Ergodic Inference

Ergodic inference (EI) is motivated by the well-known convergence of MCMC chains (Robert & Casella, 2005): MCMC chains converge in terms of the total variation (TV) distance between the marginal distribution of the MCMC chain and the target $\pi$. Inspired by the convergence property of MCMC chains, we define an ergodic approximation $P_T$ to $\pi$ with $T$ MCMC steps as following. Given a parametric distribution $P_0$ with tractable density $p_0(\mathbf{x}_0; \boldsymbol{\phi}_0)$ parameterlized by $\boldsymbol{\phi}_0$ and an MCMC kernel $M(\mathbf{x}'|\mathbf{x}; \boldsymbol{\phi})$ constructed using the unnormalised target density $\pi^*$ and with MCMC hyperparameter $\boldsymbol{\phi}$, an ergodic approximation of $\pi$ is the marginal distribution of the final state of an $T$-step MCMC chain initialized from $P_0$:

$$p_T(\mathbf{x}_T; \boldsymbol{\phi}_0, \boldsymbol{\phi}) = \int \prod_{t=1}^{T} M(\mathbf{x}_t|\mathbf{x}_{t-1}; \boldsymbol{\phi}) p_0(\mathbf{x}_0; \boldsymbol{\phi}_0) d\mathbf{x}_{0:T-1}. \tag{3}$$

We call $\boldsymbol{\phi}_0$ and $\boldsymbol{\phi}$ the ergodic parameters of $P_T$. Well known in MCMC literature like (Robert & Casella, 2005; Murray & Salakhutdinov, 2008), the ergodic approximation $p_T$ converges to $\pi$ after every MCMC transition and with sufficiently long chain $p_T$ is guaranteed to be arbitrarily close to $\pi$ with arbitrary $\boldsymbol{\phi}$ and $\boldsymbol{\phi}_0$.

It is important to clarify that ergodic approximation is different from the modified variational methods like (Ruiz & Titsias, 2019) which only optimise the variational parameters $\boldsymbol{\phi}_0$, but the optimisation objective functions involve MCMC similation. In the following section, we show how EI can tune the ergodic parameters to minimise the bias of $P_T$ as an approximation to the target $\pi$ with finite $T$.

### 3.1 ERGODIC APPROXIMATION OBJECTIVE

To reduce the KL divergence $D_{\mathrm{KL}}(P_T \parallel \pi)$, one could tune the burn-in parameter $\phi_0$ and the MCMC parameter $\phi$ by minimizing equation 2. However, this is infeasible because we cannot analytically evaluate $p_T$ in equation 3. Instead, we exploit the convergence of ergodic Markov chains and propose to optimise an alternative objective as the following constrained optimisation problem:

$$\max_{\phi_0, \phi} \quad \mathbb{E}_{p_T}[\log \pi^*(\mathbf{x})] + L_{\mathrm{ELBO}}(P_0 \parallel \pi^*) \equiv \mathcal{L}(\phi_0, \phi, \pi^*) \tag{4}$$

$$\text{subject to} \quad H(P_0) > h, \tag{5}$$

where $h$ is a hyperparameter that should be close to the entropy of the target, that is, $h \approx H(\pi)$. We call the objective in equation 4 the ergodic modified lower bound (EMLBO), denoted by $\mathcal{L}(\phi_0, \phi, \pi^*)$. Note that the EMLBO is similar to $L_{\mathrm{ELBO}}(P_T \parallel \pi^*)$, with the intractable entropy $H(P_T)$ replaced by the tractable $L_{\mathrm{ELBO}}(P_0 \parallel \pi^*)$. We now give some motivation for this constrained objective.

First, we explain the inclusion of the term $L_{\mathrm{ELBO}}(P_0 \parallel \pi^*)$ in equation 4 and its connection to $H(P_T)$. If we maximised only the first term $\mathbb{E}_p[\log \pi^*(\mathbf{x})]$ with respect to a fully flexible distribution $P$, the result would be a point probability mass at the mode of the target $\pi$. This degenerate solution is avoided in VI by optimising the sum of $\mathbb{E}_p[\log \pi^*(\mathbf{x})]$ and the entropy term $H(P)$, which enforces $P$ to move away from a point probability mass. However, $H(P_T)$ is intractable in ergodic approximation. Fotunately, we notice that maximising the term $L_{\mathrm{ELBO}}(P_0 \parallel \pi^*) = \mathbb{E}_{p_0}[\log \pi^*(\mathbf{x})] + H(P_0)$ has similar effect of maximising $H(P_T)$ for preventing $P_0$ from collapsing to the mode of $\pi$. It is easy to show that $P_T$ cannot be a delta unless $H(P_T) = -\infty$, which also implies $L_{\mathrm{ELBO}}(P_T \parallel \pi^*)$ does not exist. Since the KL divergence $D_{\mathrm{KL}}(P_t \parallel \pi)$ never increases after each MCMC transition step (Murray & Salakhutdinov, 2008), $L_{\mathrm{ELBO}}(P_t \parallel \pi^*) \geq L_{\mathrm{ELBO}}(P_{t-1} \parallel \pi^*) \geq L_{\mathrm{ELBO}}(P_0 \parallel \pi^*)$, by maximising $L_{\mathrm{ELBO}}(P_0 \parallel \pi^*)$, we also maximise $L_{\mathrm{ELBO}}(P_T \parallel \pi^*)$, which implies $L_{\mathrm{ELBO}}(P_T \parallel \pi^*)$ must exist.

The constraint in equation 5 is necessary to eliminate the following pathology. If $P_0$ does not satisfy $H(P_0) > H(\pi)$, then $\mathbb{E}_{p_0}[\log \pi^*(\mathbf{x})]$ could be higher than $\mathbb{E}_\pi[\log \pi^*(\mathbf{x})]$. When this happens, if we maximise $\mathbb{E}_{p_T}[\log \pi^*(\mathbf{x})]$, we will favor $P_T$ to stay close to $P_0$ instead of making it converge to $\pi$ faster. This is illustrated by the plot in the right part of Figure 2. To avoid this pathological case, note that $L_{\mathrm{ELBO}}(\pi \parallel \pi^*) > L_{\mathrm{ELBO}}(P_0 \parallel \pi^*)$ leads to $\mathbb{E}_\pi[\log \pi^*(\mathbf{x})] - \mathbb{E}_{p_0}[\log \pi^*(\mathbf{x})] > H(P_0) - H(\pi)$. Therefore, when $H(P_0) > H(\pi)$ is satisfied, we have that $\mathbb{E}_{p_\infty}[\log \pi^*(\mathbf{x})] = \mathbb{E}_\pi[\log \pi^*(\mathbf{x})] > \mathbb{E}_{p_0}[\log \pi^*(\mathbf{x})]$ and maximising $\mathbb{E}_{p_T}[\log \pi^*(\mathbf{x})]$ in equation 4 is expected to accelerate convergence.

It is interesting to compare the EMLBO with the objective function optimised by Salimans et al. (2015), that is, the ELBO given by

$$L_{\mathrm{ELBO}}(P_T \parallel \pi^*) - D_{\mathrm{KL}}(p(\mathbf{x}_{0:T-1}|\mathbf{x}_T) \parallel r(\mathbf{x}_{0:T-1}|\mathbf{x}_T)), \tag{6}$$

where $p(\mathbf{x}_{0:T-1}|\mathbf{x}_T)$ denotes the conditional density of the first $T$ states of the MCMC chain given the last one $\mathbf{x}_T$ and $r(\mathbf{x}_{0:T-1}|\mathbf{x}_T)$ is an auxiliary variational distribution that approximates $p(\mathbf{x}_{0:T-1}|\mathbf{x}_T)$. Note that the negative KL term in equation 6 will increase as $T$ increases. This makes the ELBO in equation 6 become looser and looser as the chain length increases. In this case, the optimisation of equation 6 results in an MCMC sampler that fits well the biased inverse model $r(\mathbf{x}_{0:T-1}|\mathbf{x}_T)$ but whose marginal distribution for $\mathbf{x}_T$ does not approximate $\pi$ well. This limits the effectiveness of this method in chains with multiple MCMC transitions. By contrast, the EMLBO does not have this problem and its optimisation will produce a more and more accurate $P_T$ as $T$ increases.

EI combines the benefits of MCMC and VI and avoids their drawbacks, as shown in Table 1. In particular, the bias in EI is reduced by using longer chains, as in MCMC, and EI generates independent samples, as in VI. Futhermore, EI optimises an objective that directly quantifies the bias of the generated samples, as in VI. Methods for tuning MCMC do not satisfy the latter and optimise instead indirect proxies for mixing speed, e.g. expected squared jumped distance (Levy et al., 2018). Importantly, EI can use gradients to tune different MCMC parameters at each step of the chain, as suggested by Salimans et al. (2015). This gives EI an extra flexibility which existing MCMC methods do not have. Finally, EI is different from parallel-chain MCMC: while EI generates independent samples, parallel-chain MCMC draws correlated samples from several chains running in parallel.

### 3.2 STOCHASTIC GRADIENT OPTIMISATION FOR THE ERGODIC OBJECTIVE

We now show how to maximise the ergodic objective using gradient-based optimisation. The gradient $\partial_{\phi_0, \phi}\mathcal{L}(\phi_0, \phi, \pi^*)$ is equal to the sum of two gradient terms. The first one $\partial_{\phi_0} L_{\mathrm{ELBO}}(P_0 \parallel \pi^*)$ is

| Method | Optimises objective quantifying bias of samples | Asymptotically unbiased | Samples generated |
|---|---|---|---|
| MCMC | No | Yes | Correlated |
| VI | Yes | No | Independent |
| EI | Yes | Yes | Independent |

Table 1: Ergodic inference combines with pros of MCMC and VI and avoids their cons.

affected by the constraint $H(P_0) > h$, while the second term $\partial_\phi \mathbb{E}_{p_T}[\log \pi^*(\mathbf{x}_T)]$ is not. If we ignore the constraint, the first gradient term can be estimated by Monte Carlo using the reparameterization trick proposed in (D.P. Kingma, 2014; Rezende & Mohamed, 2015):

$$\partial_{\phi_0} L_{\text{ELBO}}(P_0 \parallel \pi^*) \approx \frac{1}{N} \sum_{i=1}^{N} \partial_{\phi_0} \log \pi^*(f_{\phi_0}(\boldsymbol{\epsilon}_i)) + \partial_{\phi_0} H(P_0), \qquad (7)$$

where $f_{\phi_0}(\cdot)$ is a deterministic function that maps the random variable $\boldsymbol{\epsilon}_i$ sampled from a simple distribution, e.g. a factorized standard Gaussian, into the random variable $\mathbf{x}_0^i$ sampled from $p_0(\cdot; \boldsymbol{\phi}_0)$. To guarantee that our gradient-based optimiser yields a solution satisfying the constraint, we first initialize $\boldsymbol{\phi}_0$ so that $H(P_0) > h$ and, afterwards, we force the gradient descent optimiser to leave $\boldsymbol{\phi}_0$ unchanged if $H(P_0)$ is to get lower than $h$ during the optimisation process.

The Monte Carlo estimation of $\partial_\phi \mathbb{E}_{p_T}[\log \pi^*(\mathbf{x})]$ can also be computed using the reparameterization trick. For this, the Metropolis-Hastings (M-H) correction step in the MCMC transitions, as described in Section 2.2, can be reformulated as applying the following transformation to $\mathbf{x}_{t-1}$:

$$\mathbf{x}_t = g_\phi(\mathbf{x}_{t-1}, \mathbf{r}_t; u_t) = \mathbf{x}' \mathbf{1}(p_{\text{MH}}; u_t) + \mathbf{x}_{t-1}[1 - \mathbf{1}(p_{\text{MH}}; u_t)], \qquad (8)$$

where $(\mathbf{x}', \mathbf{r}') = f_\phi(\mathbf{x}_{t-1}, \mathbf{r}_t)$ as described in Section 2.2, $\mathbf{r}_t \sim q(\mathbf{r})$, $u_t \sim \text{Unif}(0, 1)$, $p_{\text{MH}} = \min\{1, \pi^*(\mathbf{x}')q(\mathbf{r}')/[\pi^*(\mathbf{x}_{t-1})q(\mathbf{r}_t)]\}$ and $\mathbf{1}(p_{\text{MH}}; u)$ is an indicator function that takes value one if $p_{\text{MH}} > u$ and zero otherwise. In Hamiltonian Monte Carlo (HMC), $f_\phi$ is the leapfrog integrator of Hamiltonian dynamics with the leapfrog step size $\phi$. We define the $T$-times composition of $g_\phi$, given in equation 8, as the transformation $\mathbf{x}_T = g_\phi^T(\mathbf{x}_0, \mathbf{r}_{1:T}; u_{1:T})$. Then, the second gradient term can be estimated by Monte Carlo as follows:

$$\partial_\phi \mathbb{E}_{p_T}[\log \pi^*(\mathbf{x}_T)] \approx \frac{1}{N} \sum_{i=1}^{N} \partial_\phi \log \pi^*[g_\phi^T(\mathbf{x}_0^i, \mathbf{r}_{1:T}^i; u_{1:T}^i)], \qquad (9)$$

where $\mathbf{x}_0^i$, $\mathbf{r}_{1:T}^i$ and $u_{1:T}^i$ are sampled independently from $p_0(\mathbf{x}_0) \prod_{t=1}^{T} q(\mathbf{r}_t) \text{Unif}(u_t; 0, 1)$. Note that the gradient term equation 9 is correct under the assumption $f_\phi$ is volume-preserving in the joint space of $(\mathbf{x}_{t-1}, \mathbf{r}_t)$, otherwise additional gradient term of the Jacobian of $f_\phi$ w.r.t. $\phi$ is required. However, it is not a concern for many popular MCMC kernels. For example, the leapfrog integrator in HMC $f_\phi$ guarantees the preservation of volume as shown in (Neal, 2010). It is worth to mention that the indicator function in equation 8 is not continuous but differentiable almost everywhere. Therefore, the gradient in equation 9 can be computed conveniently using standard autodifferentiation tools.

The gradient in equation 9 requires computing $\partial_\phi g_\phi^T(\mathbf{x}_0, \mathbf{r}_{1:T}; u_{1:T})$, which can be done easily by using auto-differentiation and gradient backpropagation through the transfromations $g_\phi(\cdot, \mathbf{r}_t; u_t)$ with $t = T, \ldots, 1$. However, backpropagation in deep compositions can be computationally demanding. We discovered a trick to accelerate the gradient computation by stopping the backpropagation of the gradient at the input $\mathbf{x}_{t-1}$ of $g_\phi(\mathbf{x}_{t-1}, \mathbf{r}_t; u_t)$, for $t = 1, \ldots, T$. Empirically this trick has almost no impact on the convergence speed of the ergodic approximation, as shown in Figure 2.

### 3.3 THE ENTROPY CONSTRAINT AND HYPERPARAMETER TUNING

As mentioned previously, ignoring the constraint $H(P_0) > H(\pi)$ may lead to pathological results when optimising the ergodic objective. To illustrate this, we consider fitting an ergodic approximation given by a Hamilton Monte Carlo (HMC) transition kernel with $T = 9$. $P_9$ denotes the initial ergodic approximation before traing and $P_9^*$ denotes the same approximation after training. The target distribution is a correlated bivariate Gaussian given by $\pi = \mathcal{N}(\mathbf{0}, (2.0, 1.5; 1.5, 1.6))$. Samples

from this distribution are shown in plot (a) in Figure 1. We optimise different a separate HMC parameter $\phi_t$, as described in Section 2.2, for each HMC step $t$. We consider two initial distributions. The first one is $P_0 = \mathcal{N}(\mathbf{0}, 3\mathbf{I})$ which satisfies the assumption $H(P_0) > H(\pi)$. The second one is $P_0' = \mathcal{N}(\mathbf{0}, \mathbf{I})$ with the entropy $H(P_0') < H(\pi)$, which violates the assumption. In this latter case, we perform the unconstrained optimisation of equation 4. Plots (b) and (c) in Figure 1 show samples from $P_9$ and $P_9^*$ for the valid $P_0$. In this first example, maximising the ergodic objective under equation 5 significantly accelerates the chain convergence as further shown by the left plot in Figure 2. Plots (d) and (e) in Figure 1 show samples from $P_9$ and $P_9^*$ for the invalid initial distribution $P_0'$. In the second example, $\mathbb{E}_{p_0}[\log \pi^*(\mathbf{x})]$ is higher than $\mathbb{E}_{\pi}[\log \pi^*(\mathbf{x})]$ and, consequently, maximising the unconstrained ergodic objective actually deteriorates the quality of the resulting approximation. This is further illustrated by the right plot in Figure 2 which shows how the convergence of $\mathbb{E}_{p_t}[\log \pi^*(\mathbf{x})]$ to $\mathbb{E}_{\pi}[\log \pi^*(\mathbf{x})]$ is significantly slowed down by the optimisation under the invalid $P_0'$.

Fortunately, it is straightforward to prevent this type of failure cases by appropriately tuning the scalar hyperparameter $h$ in equation 5. A value of $h$ that is too low may result in higher bias of $P_T$ after optimisation as illustrated by the convergence of $\mathbb{E}_{p_t}[\log \pi^*(\mathbf{x})]$ in the blue and orange curves in Plot (b) in Figure 2. Furthermore, in many cases, estimating an upper bound on $H(\pi)$ is feasible. For example, in Bayesian inference, the entropy of the prior distribution $p(\mathbf{x})$ is often higher than the entropy of the posterior $p(\mathbf{x}|\mathbf{y})$. Therefore, the prior entropy can be used as a reference for tuning $h$.



(a) $\pi$      (b) $P_9$ with $P_0$      (c) $P_9^*$ with $P_0$      (d) $P_9$ with $P_0'$      (e) $P_9^*$ with $P_0'$

Figure 1: Histograms of samples from ergodic inference using HMC transition kernels. $P_9$ denotes the ergodic approximation before traing; $P_9^*$ denotes the ergodic approximation after training.

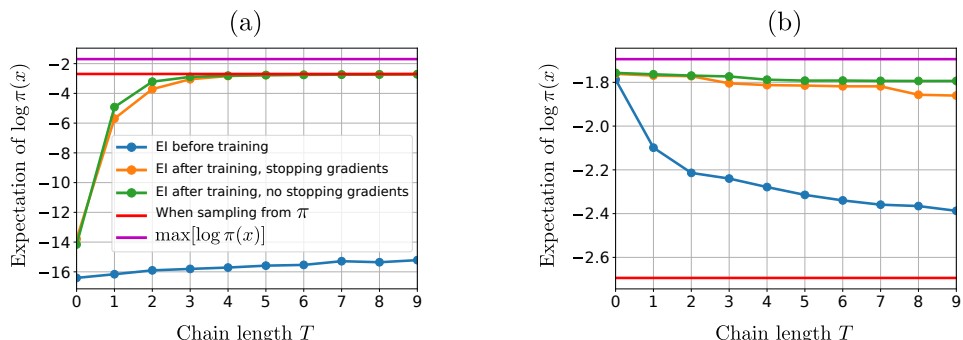

Figure 2: The plot of $\mathbb{E}_{p_T}[\log \pi^*(\mathbf{x})]$ as a function of the length of the chain $T$ using 10000 samples: Left: with the valid $P_0$ as $H(P_0) > H(\pi)$; Right: with invalid $P_0'$ as $H(P_0') < H(\pi)$. SG training means the stop gradient is applied to the $\mathbf{x}$ from previous HMC step in equation 9.

## 4 EXPERIMENTS

We first describe the general configuration of the ergodic inference method used in our experiments. Our ergodic approximation is constructed using HMC, one of the most successful MCMC methods in machine learning literature. We use $T$ HMC transitions, each one involving 5 steps of the vanilla leapfrog integrator which was implemented following Neal (2010). The leapfrog pseudocode can be found in the appendix. In each HMC transition, the auxiliary variables are sampled from a zero-mean Gaussian distribution with diagonal covariance matrix. We tune the following HMC parameters: the variance of the auxiliary variables and the leapfrog step size, as mentioned in Section 2.2. We use and optimise a different value of the HMC parameters for each of the $T$ HMC transitions considered. We call our ergodic inference method Hamiltonian ergodic inference (HEI). The burn-in model $P_0$ is

factorized Gaussian. The initial entropy of $P_0$ is chosen to be the same as the entropy of the prior. The stocastic optimisation algorithm is Adam (Kingma & Ba, 2015) with TensorFlow implemtation Abadi et al. (2015) and the optimiser hyperparameter setting is ($\beta_1 = 0.9, \beta_2 = 0.999, \epsilon = 10^{-8}$). The initial HMC leapfrog step sizes are sampled uniformly between 0.01 and 0.025. Additional experiment on Bayesian neural networks is included in Appendix 6.3.

## 4.1 DEMONSTRATIONS

We first compare Hamiltonian ergodic inference (HEI) with previous related methods on 6 synthetic bivariate benchmark distributions. Histograms of ground truth samples from each target distribution using rejection sampling are shown in Figure 3. The baselines considered include: 1) Hamiltonian variational inference (HVI) (Salimans et al., 2015); 2) generalized Hamiltonian Monte Carlo (GHMC) using an NVP parameterized HMC kernel and gradient-based auto-tuning of MCMC parameters w.r.t. sample correlation loss (Levy et al., 2018); 3) Hamiltonian annealed importance sampling (HAIS) (Sohl-Dickstein & Culpepper, 2012). It is worth to mention that we do not consider other hybrid inference methods like (Ruiz & Titsias, 2019; Hoffman, 2017) in our experiment, because these methods only combines MCMC simulation with VI but not optimise the parameters of MCMC kernel using the gradient-based approach like EI.

HVI is the most similar method to HEI among all three baselines, because both HEI and HVI methods generate samples from the last state of MCMC chains and use gradient-based MCMC hyperparameter tuning to reduce bias. For a fair comparison between HVI and HEI, we consider the HMC chains with exactly the same setting in both methods: the initial state follows a standard Gaussian distribution and the length of HMC chain is $T = 10$. The key difference between HVI and HEI is the hyperparameter tuning objective, as mentioned in Section 3.1. We trained HVI for 1000 iterations and verified the ELBO converges to a (local) minimum (plots of the training ELBO values are included in Appendix 6.2). We trained HEI for 50 iterations. Following the setting of HAIS by Wu et al. (2017), we used 1,000 intermediate distributions with 5 leapfrog steps per HMC transition and manually tuned the HMC parameters to have acceptance rate around 70%. GHMC[1] was run using 100 parallel chains with 5 leapfrog steps per GHMC transition, 100 burn-in steps and 1000 auto-tuned training iterations Levy et al. (2018). The verification of the convergence of $\mathbb{E}_{p_T}[\log \pi^*(x)]$ to $\mathbb{E}_\pi[\log \pi^*(x)]$ for HEI is shown in plot (a) of Figure 5.

We generate 100,000 samples with each method and evaluate sample quality using two metrics: 1) the histogram of simulated samples for visual inspection; 2) the MC estimation of $\mathbb{E}_\pi[\log \pi^*(\mathbf{x})]$. Effective sample size (ESS) is a popular sample correlation based evaluation metric in recent MCMC literature (Levy et al., 2018). However, we do not consider ESS in this experiment, because GHMC is the only method among all methods generating correlated samples. Therefore, the ESS of GHMC is guaranteed to be lower than HVI and HEI. To generate ground truth samples from benchmark distributions, we use . The resulting sample histograms of the ground truth using rejection sampling are shown in figures 3 and considered approximated sampling methods are shown in 4. Table 2 shows the resulting estimates of $-\mathbb{E}_\pi[\log \pi^*(\mathbf{x})]$ together with the wall-clock simulation time for generating 100,000 samples. The left part of Table 3 shows the training time of the MCMC parameter optimisation for all methods except HAIS, which does not support gradient-based HMC hyperparameter tuning. HEI is faster than HVI and GHMC. Note, however, that the acceleration of HEI over HSVI is due to the stopping gradient trick described in Section 3.2. The histograms and the estimates of $-\mathbb{E}_\pi[\log \pi^*(\mathbf{x})]$ generated by HEI are consistent with the results of the more expensive unbiased samplers GHMC and HAIS, which are close to the ground truth. By contrast, HVI exhibits a clear bias in all benchmarks. Regarding the sampling time, HVI and HEI simulate HMC chains with the same length and, consequently, perform similarly in this case while sample simulation from HAIS and GHMC is much more expensive.

---

[1]The code used was obtained from https://github.com/brain-research/l2hmc

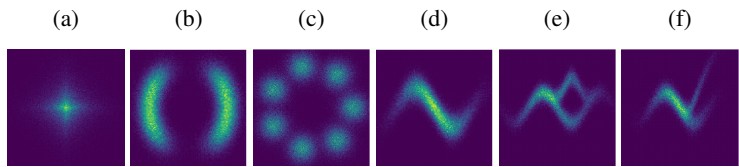

Figure 3: Histograms of samples generated by rejection sampling on each benchmark problem.

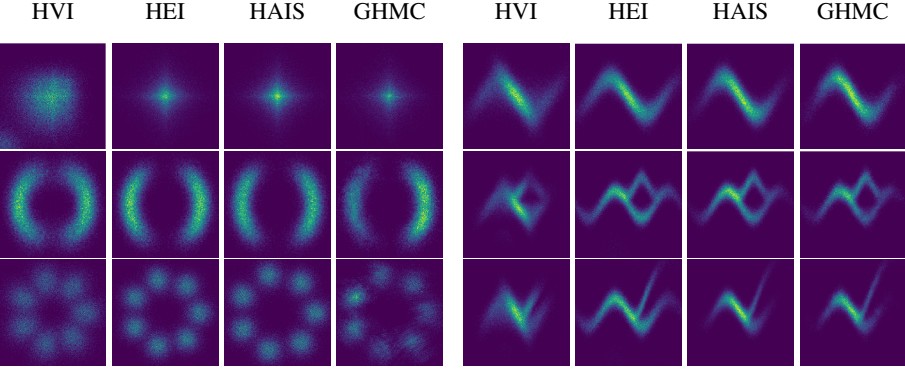

Figure 4: Histograms of 100,000 samples generated by each method after parameter optimisation.

| Encoders | Training hours | Training Epochs | $\log(\mathbf{x})$ | Test ESS |
|---|---|---|---|---|
| Conv VAE($n_h$=300) Salimans et al. (2015) | - | - | -83.20 | - |
| HVI($T$=1, 16LF, $n_h$=800, ConvNet encoder) Salimans et al. (2015) | - | - | -81.94 | - |
| HVAE($T$=1, 20LF, $n_h = 300$, ConvNet encoder) Caterini et al. (2018) | - | - | -84.78 | - |
| Conv VAE($n_h$=500) (**Baseline**) | 6.00 | 3000 | -83.57 | 50 |
| HVI($T$=1, 16LF, $n_h$=800, ConvNet encoder) | 6.00 | 360 | -83.68 | 48 |
| HVAE($T$=1, 16LF, $n_h$=500, ConvNet encoder) | 6.00 | 360 | -84.22 | 48 |
| HEI($T$=30, 5LF, $n_h$=500, no neural net encoder) | 1.65 | 54 | -83.17 | 48 |
| HEI($T$=30, 5LF, $n_h$=500, no neural net encoder) | 3.00 | 100 | -82.76 | 46 |
| HEI($T$=30, 5LF, $n_h$=500, no neural net encoder) | 6.00 | 200 | -82.65 | 45 |
| HEI($T$=30, 5LF, $n_h$=500, no neural net encoder) | 12.00 | 400 | **-81.43** | 38 |
| HEI($T$=15, 5LF, $n_h$=500, no neural net encoder) | 8.00 | 540 | -83.30 | 48 |

Table 4: Comparisons in terms of compuational efficiency and test log-likelihood in the training of deep generative models on the MNIST dataset. We implemented the deconvolutional decoder network in Salimans et al. (2015) to test HVI. In Salimans et al. (2015), the test likelihood is estimated using importence-weighted samples from the encoder network. In our experiment, we use Hamiltonian annealed importance sampling and report the effective sample size (ESS).

## 4.2 TRAINING DEEP GENERATIVE MODELS

We now evaluate HEI in the task of training deep generative models. MNIST is a standard benchmark problem in this case with 60,000 grey level $28 \times 28$ images of handwritten digits. For fair comparison with previous works, we use the 10,000 prebinarised MNIST test images[2] used by Burda et al. (2015). The architecture of the generative model considered follows the deconvolutional network from Salimans et al. (2015). In particular, the unnormalised target $p_{\boldsymbol{\theta}}(\mathbf{x}, \mathbf{y})$ consists of 32 dimensional latent variables $\mathbf{x}$ with Gaussian prior $p(\mathbf{x}) = \mathcal{N}(\mathbf{0}, \mathbf{I})$ and a deconvolutional network $p_{\boldsymbol{\theta}}(\mathbf{y}|\mathbf{x})$ from top to bottom including a single fully-connected layer with 500 RELU hidden units, then three deconvolutional layers with $5 \times 5$ filters, (16, 32, 32) feature maps, RELU activations and a logistic output layer. We consider a baseline given by a standard VAE with a factorised Gaussian approximate

---

[2]https://github.com/yburda/iwae

| $-\mathbb{E}_\pi[\log \pi^*(\mathbf{x})]$ estimate / time | a | b | c | d | e | f |
|---|---|---|---|---|---|---|
| HVI Salimans et al. (2015) | 4.94/0.20 | 1.22/0.41 | 5.22/0.92 | 1.70/0.25 | 1.37/0.87 | 1.64/0.54 |
| HEI | 3.49/0.25 | 0.79/0.44 | 4.78/1.58 | 0.99/0.3 | 0.59/0.58 | 0.57/0.56 |
| GHMC Neal (2001) | 3.43/35.7 | 0.80/64.28 | 4.74/41.23 | 1.03/48.54 | 0.63/85.30 | 0.59/81.0 |
| HAIS Levy et al. (2018) | 3.38/16.62 | 0.78/26.94 | 4.70/125.32 | 1.00/22.61 | 0.60/33.07 | 0.49/33.69 |
| Ground Truth | 3.31/- | 0.76/- | 4.66/- | 0.98/- | 0.58/- | 0.49/- |

Table 2: Estimation of $-\mathbb{E}_\pi[\log \pi^*(\mathbf{x})]$ and the sampling time on CPU: Each score (a/b) above refers to: a) $-\mathbb{E}_\pi[\log \pi^*(\mathbf{x})]$ estimated by 100k samples; b) time in seconds to generate 100,000 samples.

| Training Time on Synthetic Problems | |
|---|---|
| **Method** | **sec / 100 iters** |
| HVI (Salimans et al., 2015) | 2.367 |
| HEI (stop gradient) | 1.620 |
| GHMC (Neal, 2001) | 7.100 |

| Training Time on DGM (sec / epoch) | | |
|---|---|---|
| **Method** | **T=15** | **T=30** |
| HEI | 257.8 | 551.2 |
| HEI (stop gradient) | 56.3 | 114.0 |

Table 3: Left. The training time of MCMC parameter optimisation in seconds for 100 iterations for all candidate methods to produce the results in Figure 4. The training time of HEI is lower than HVI because of the stop gradient trick mentioned in Section 3.2. We do not report the training time for HAIS, because HAIS requires manual tuning of MCMC hyperparameters which is not directly comparable to the gradient-based autotuning used by the other methods. Right. The training time in seconds per epoch for the experiments with deep generative models (DGM).

posterior generated by an encoder network $q(\mathbf{x}|\mathbf{y})$ which mirrors the architecture of the decoder (Salimans et al., 2015).

The code for HVI Salimans et al. (2015) is not publicly available. Nevertheless, we reimplemented their convolutional VAE and were able to reproduce the marginal likelihood reported by Salimans et al. (2015), as shown in Table 4. This verifies that our implementation of the generation network is correct. We implemented HVI in (Salimans et al., 2015) using an auxiliary reverse model in the ELBO parameterized by a single hidden layer network with 640 hidden units and RELU activations. We also implemented the Hamiltonian variational encoder (HVAE) method (Caterini et al., 2018), which is similar to HVI but without the reverse model. Unlike in the original HVAE, our implementation does not use tempering but still produces results similar to those from Caterini et al. (2018).

For the HEI encoder, we use $T = 30$ HMC steps, each with 5 leapfrog steps. The initial approximation $P_0$ is kept fixed to be the prior $p(\mathbf{x})$. We optimise the decoder and the HEI encoder jointly using Adam. Table 4 shows the marginal test log-likelihood for HEI and the other methods, as estimated with 1,000 HAIS samples (Sohl-Dickstein & Culpepper, 2012). Following Li et al. (2017), we also include the effective sample size (ESS) of HAIS samples for the purpose of verifying the reliability of the reported test log-likelihoods. Overall, HEI outperforms HVI, HVAE and the standard VAE in test log-likelihood when the training time of all methods is fixed to be 6 hours. HEI still produces significant gains when the training time is extended to 12 hours and, with only 1.6 hours of training, HEI can already outperform the convolutional VAE of Salimans et al. (2015) with 6 hours of training.

To verify the convergence of HEI, we show in plot (b) of Figure 5 estimates of $\mathbb{E}_{p_T}[\log \pi^*(\mathbf{x})] - \mathbb{E}_\pi[\log \pi^*(\mathbf{x})]$ for $T = 1, \ldots, 10$ on five randomly chosen test images, where the ground truth $\mathbb{E}_\pi[\log \pi^*(\mathbf{x})]$ is estimated by HAIS, after HMC hyper-parameter tuning in HEI (blue) and without hyper-parameter tuning in HEI (green), i.e. just using the initial hyper-parameter values. Plot (c) in Figure 5 shows similar results, but using the maximum mean discrepancy (MMD) score (Gretton et al., 2012) to quantify the similarity of samples from $p_T$ to samples from $\pi$, where the latter ground truth samples are generated by HAIS. These plots suggests that shortening the HEI chain to $T = 10$ HMC steps will have a negligible effect on final simulation accuracy. Finally, the right part of Table 3 shows the training time of HEI with and without the stopping gradient trick. These resuls show that the former method is up to 5 times faster.

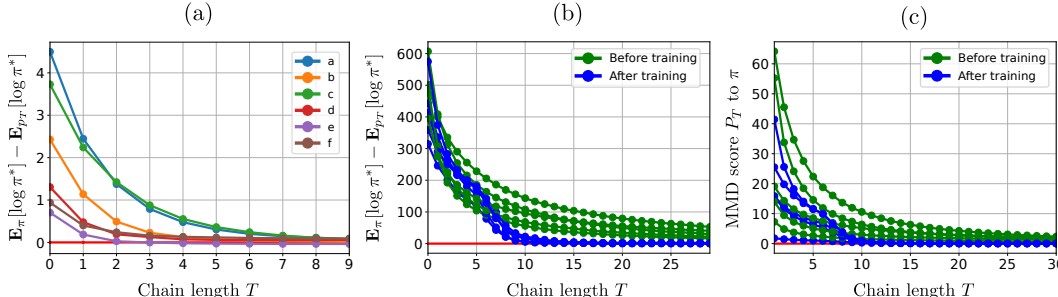

Figure 5: The verification of the convergence of $\mathbb{E}_{p_T}[\log \pi^*(\mathbf{x})]$ to $\mathbb{E}_\pi[\log \pi^*(\mathbf{x})]$: a: the targets are 2D benchmarks with the ground truth of $\mathbb{E}_\pi[\log \pi^*(\mathbf{x})]$; b: the target $\pi$ is the VAE posterior $p(\mathbf{x}|\mathbf{y})$ each curve represents one random chosen test MNIST image $\mathbf{y}$ with the ground truth of $\mathbb{E}_\pi[\log \pi^*(\mathbf{x})]$ estimated by HAIS using 100 samples; c: MMD score between HEI samples and HAIS samples.

## 5 SUMMARY

We have proposed Ergodic Inference (EI), a novel hybrid inference method that bridges MCMC and VI. EI a) reduces the approximation bias by increasing the number of MCMC steps, b) generates independent samples and c) tunes MCMC hyperparameters by optimising an objective function that directly quantifies the bias of the resulting samples. The effectiveness of EI was verified on synthetic examples and on popular benchmarks for deep generative models. We have shown that we can generate samples much closer to a gold standard sampling method than similar hybrid inference methods and at a low computational cost. However, one disadvantage of EI is that it requires the entropy of the first MCMC step to be larger than the entropy of the target distribution.

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

## 6 APPENDIX

### 6.1 LEAPFROG ALGORITHM

Here is the code for the vanilla leapfrog algorithm we used in HVI, HEI and HAIS.

---
**Algorithm 1:** Leapfrog

---
**Input:** $\mathbf{x}$: state, $\mathbf{r}$: momenta, $\phi_1$: $\mathbf{r}$ variance, $\phi_2$: step size, $m$: number of steps
**Result:** $\mathbf{x}'$: new state, $\mathbf{r}'$: new momentum
$\bar{\mathbf{x}} = \mathbf{x}$;
$\bar{\mathbf{r}} = \mathbf{r}$;
**for** $t \leftarrow 1$ *to* $m$ **do**
$\quad$ $\bar{\mathbf{r}} = \bar{\mathbf{r}} - 0.5\phi_2\partial_{\mathbf{x}}U(\bar{\mathbf{x}})$;
$\quad$ $\bar{\mathbf{x}} = \bar{\mathbf{x}} + \phi_2/\phi_1\bar{\mathbf{r}}$;
$\quad$ $\bar{\mathbf{r}} = \bar{\mathbf{r}} - 0.5\phi_2\partial_{\mathbf{x}}U(\bar{\mathbf{x}})$;
**end**
$\mathbf{x}' = \bar{\mathbf{x}}$;
$\mathbf{r}' = \bar{\mathbf{r}}$;
**return** $\mathbf{x}'$ and $\mathbf{r}'$;

---

### 6.2 RESULTS OF HVI ON SYNTHETIC BENCHMARKS

The plots in Figure 6 show training loss (negative ELBO) of HVI and the training expected log likelihood $\mathbb{E}_{p_T}[\log \pi^*(\mathbf{x})]$ with $T = 10$ HMC steps with Adam with hyperparameter setting described in Section 4. It is clear that HVI is well trained but the approximation is biased, because $\mathbb{E}_{p_T}[\log \pi^*(\mathbf{x})]$ does not converge to the true loss (the red line on the right plots). In comparison, in Figure 6(Left) in our paper, $\mathbb{E}_{p_T}[\log \pi^*(\mathbf{x})]$ of HEI converges to the ground true by optimising our ergodic loss.

### 6.3 BAYESIAN INFERENCE WITH NEURAL NETWORKS

In this additional experiment we approximate the posterior distribution of Bayesian neural networks with standard Gaussian priors. We consider four UCI datasets and compare HEI with the stochastic gradient Hamilton Monte Carlo (SGHMC) method from Springenberg et al. (2016). The networks used in this experiment have 50 hidden layers and 1 real valued output unit, as stated in Springenberg et al. (2016). The HEI chain contains 50 HMC transformation with 3 Leapfrog steps each. The initial proposal distribution $P_0$ is a factorised Gaussian distribution with mean values obtained by running standard mean-field VI using Adam for 200 iterations. We do not use in $P_0$ the variance values returned by VI because these are unlikely to result in higher entropy than the exact posterior since VI tends to understimate uncertainty. Instead, we choose the marginal variances to be $n^{-0.5}$ where $n$ is the number of inputs to the neural network layer for the weight. To reduce computational cost, we use in this case stochastic gradients in the leapfrog integrator. For this, we split the training data into 19 mini-batches and only use one random sampled mini-batch for computing the gradient in each leapfrog iteration. We train our HEI for 10 epochs and the stationary distribution is chosen as approximate posterior on a random sampled mini-batch. The resulting test log-likelihoods are shown in Table 5. Overall, HEI produce significantly better results than SGHMC. We also show in the right plot of Figure 7 estimates of $\mathbb{E}_{p_t}[\log p(\mathbf{x}, \mathbf{y})]$ for $t = 1, \ldots, 50$ after HMC hyper-parameter tuning and without hyper-parameter tuning.

| Method/Dataset | Boston | Yacht | Concrete | Wine |
|---|---|---|---|---|
| SGHMC (best average) Springenberg et al. (2016) | -3.47±0.51 | -13.58±0.98 | -4.87±0.05 | -1.82±0.75 |
| SGHMC (tuned per dataset) Springenberg et al. (2016) | -2.49±0.15 | -1.75±0.19 | -4.16±0.72 | -1.29±0.28 |
| SGHMC (scale-adapted) Springenberg et al. (2016) | -2.54±0.04 | -1.11±0.08 | -3.38±0.24 | -1.04±0.17 |
| HEI | **-2.17±0.07** | **-0.47±0.06** | **-2.71±0.03** | **-0.71±0.03** |

Table 5: The log likelihood on UCI datasets averaged over 20 splits.

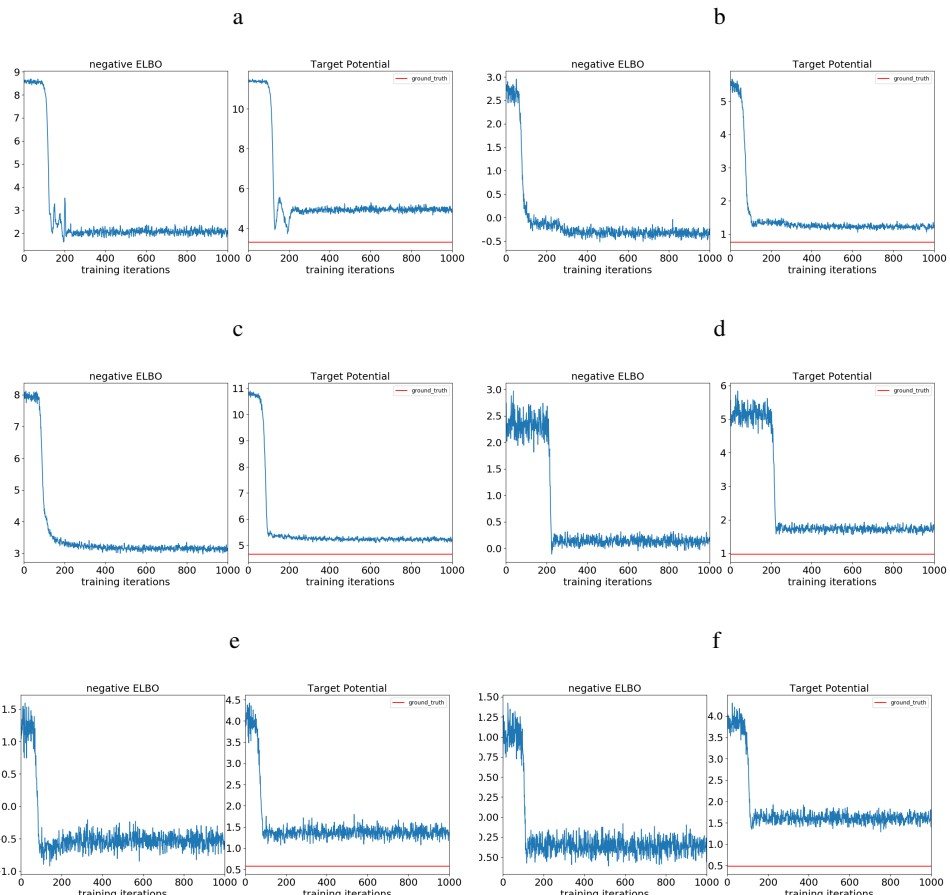

Figure 6: histograms of samples from benchmarks by rejection sampling

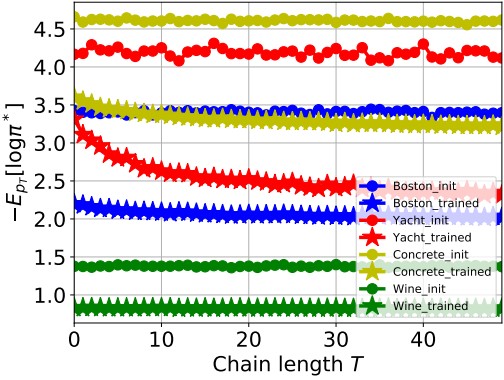

Figure 7: The convergence of sample likelihood from ergodic approximation on Bayesian NN

