# OpenReview forum: "Ergodic Inference: Accelerate Convergence by Optimisation"
_ICLR.cc/2020/Conference — Reject_

### Official Review · AnonReviewer2 · 2019-10-07
**Official Blind Review #2**

**Rating:** 3

**Review:**

This paper proposes a new combination of Markov chain Monte Carlo (MCMC) and variational inference (VI) for improving approximate inference. The main contribution is the optimization objective that allows improving the quality of samples obtained from the combination of VI and MCMC. Specifically, the authors minimize the "approximate" version of the Kullback-Leibler (KL) divergence between the distribution of MCMC + VI and the true distribution. The authors validate the effectiveness of their formulation through experiments on 6 synthetic benchmarks and generative modeling of MNIST (experiments on Bayesian neural networks are also provided in the appendix).

Overall, I think the paper provides a solid contribution towards combining MCMC and VI by proposing a way to optimize the MCMC part. The experiments validate the method by showing consistent improvement over existing methods. However, I believe the justification behind the proposed formulation, i.e., Equation (4) and (5), needs to be improved before being published at the conference.

First, for Equation (4), the explanation behind "replacing" H(P_{T}) with ELBO w.r.t. P_{0} is confusing. Specifically, it is reasoned that ELBO w.r.t. P_{t} only increase after MCMC steps. This statement is misleading since the replacement was done for H(P_{T}), not the ELBO w.r.t. P_{T}.

I also think the Equation (5) is not properly justified. it is stated that the constraint is needed for preventing P_{T} to be closer to P_{0}. However, nothing is stated about the reason on why P_{T} gets closer to \pi when Equation (5) is satisfied.  Note that even if the expected log-likelihood of the distribution is high, it does not necessarily mean that the distribution is more similar.

Minor comments:
- I was unable to understand why the algorithm is named "ergodic" inference. Both HVI and the proposed EI rely on the ergodic property of Markov chain for improving the variational distribution. I hope the authors could better illustrate on this point. I also think the term "ergodic approximation" in page 3. is hard to understand.
- I (weakly) suggest changing y-axis of Figure 5. to log-scale for better readability. It almost seems that the brown plot does not converge in Fig 5-(a).
- The paper could have been strengthened by performing experiments on more challenging datasets, e.g., CIFAR-10 or CIFAR-100.


**Experience Assessment:**

I have read many papers in this area.

**Review Assessment: Checking Correctness Of Derivations And Theory:**

I assessed the sensibility of the derivations and theory.

**Review Assessment: Checking Correctness Of Experiments:**

I assessed the sensibility of the experiments.

**Review Assessment: Thoroughness In Paper Reading:**

I read the paper at least twice and used my best judgement in assessing the paper.

---

> ### Author Response · Authors · 2019-11-15
> **Rebuttal**
>
> Thanks for your valuable feedback. We would like to address concerns and questions in the review as following:
>
> 1)"First, for Equation 4, the explanation behind "replacing" H(P_{T}) with ELBO w.r.t. P_{0} is confusing...":
>
> Yes, this statement could be confusing, we will rephrase it in a better word if the paper is accepted. However, it is important to clarify that we explained in the paper that including the ELBO w.r.t. P_{0} in ergodic objective (Equation 4) is motivated by the similar effect as including H(P_{T}) in the ELBO of P_{T}, that is stopping P_T collapsing to the mode of \pi. This is briefly explained in the paper " we instead replace this
> term with ...This also prevents PT from collapsing to the mode ... and prevent PT from collapsing to a delta." on Page 4.
>
> It is very important to know the *fact* that, if the ELBO of P_T is finite, H(P_T) cannot be minus infinity (which is equivalent to P_T collapsing to delta at the model of \pi). As mentioned in the paper, maximising ELBO of P_0 by guarantees maximising the ELBO of P_T. Therefore, maximising ELBO of P_0 guarantees that the ELBO of P_T is finite, that is H(P_T) > ELBO(P_0) - E_{P_T}[\log \pi] > -\infty.
>
>
> 2) "nothing is stated about the reason on why P_{T} gets closer to \pi when Equation (5) is satisfied.":
>
> The convergence of MCMC chain is the *reason* to why P_{T} gets closer to \pi *no matter* Equation (5) is satisfied or not. This is well known in MCMC literature [1, 2]. In particular, we mentioned the two important literature [1] and [2] in the submitted paper to clarify this. We have explicitly clarified this in the revised version.
>
> The reason to including Equation (5) is to avoid a pathological case in optimising the ergodic objective (Equation 4). The pathology has been explained in our paper, see the paragraph: "The constraint in Equation 5 is necessary to eliminate the following pathology...". To clarify this important pathology further, we dedicated Section 3.3 to demonstrate the pathological case in a correlated 2-d Gaussian example when Equation 4 is optimised without Equation 5 and showed this pathological case can be fixed by including Equation 5. This is illustrated as Figure 1 and 2.
>
> [1] Iain Murray and Ruslan Salakhutdinov. Notes on the KL-divergence between a Markov chain and its
> equilibrium distribution. preprint, 2008.
>
> [2] Christian P. Robert and George Casella. Monte Carlo Statistical Methods (Springer Texts in Statistics).
> Springer-Verlag New York, Inc., Secaucus, NJ, USA, 2005. ISBN 0387212396.
>
> 3)"I was unable to understand why the algorithm is named "ergodic" inference. Both HVI and the proposed EI rely on the ergodic property of Markov chain for improving the variational distribution. I hope the authors could better illustrate on this point. I also think the term "ergodic approximation" in page 3. is hard to understand.":
>
> The ergodicity is about convergence of MCMC chain to target distributions *as the length of the chain increases*. Ergodic inference is more than just using a few MCMC steps to reduce bias, but EI provides to *accelerate* the convergence with finite number of MCMC steps. This is why we demonstrate optimising the proposed Ergodic Objective can significantly improve the decay of bias in log likelihood and maximum mean discrepancy (MMD) score in Figure 5.
>
> We explained the key difference in the loss function between EI and HVI in the paragraph "It is interesting to compare the EMLBO with the objective function optimised by Salimans et al. (2015),...". It is important to mention that HVI and HVAE only use 1 HMC step to reduce some bias in their experiments. In contrast, EI work better in general with multiple MCMC steps. To demonstrate the advantage of Ergodic Objective over the ELBO used in HVI, in Section 4.1 we showed that EI outperforms HVI in sample bias in all 2-d benchmarks, even with exactly the same setting of HMC chains with multiple HMC steps.

---

### Official Review · AnonReviewer1 · 2019-10-19
**Official Blind Review #1**

**Rating:** 8

**Review:**

The presented method is very useful to deep learning in the era of uncertainty modelling, which requires the use of Bayesian inference arguments. It's a valuable improvement upon variational inference, it's novel, and the derivations are correct. The presentation is elaborate and covers all expected aspects. The literature review is up to date.
The experimental results are diverse enough and convincing. The authors have considered both proof of concept experiments and deep learning architectures. The comparisons are valid.


**Experience Assessment:**

I have published in this field for several years.

**Review Assessment: Checking Correctness Of Derivations And Theory:**

I carefully checked the derivations and theory.

**Review Assessment: Checking Correctness Of Experiments:**

I carefully checked the experiments.

**Review Assessment: Thoroughness In Paper Reading:**

I read the paper thoroughly.

---

### Official Review · AnonReviewer3 · 2019-10-23
**Official Blind Review #3**

**Rating:** 3

**Review:**

The paper presents a new hybrid method to unify MCMC and VI. The key idea is to interpret a ﬁnite-length MCMC/HMC chain as a parametric procedure, whose parameters can be optimized via a VI-motivated objective. Specifically, the authors propose to modify the well-known ELBO (which is now non-trivial due to the intractable entropy) to form a new constrained and tractable objective. The presented techniques are tested on synthetic datasets and with the experiments of a VAE on MNIST.

The presented technique is interesting. However, there are several concerns of mine that should be addressed, as detailed below.

The notations of \pi and \pi^* are very confusing. I guess \pi represents the marginal distribution of the last state of the MCMC chain, while \pi^* is the target distribution. Is that right? Please clarify their meanings.

There are related works that combine MCMC and VI, such as [1]. What are the advantages of the proposed method compared to that method?
[1] Francisco J. R. Ruiz and Michalis K. Titsias. A Contrastive Divergence for Combining Variational Inference and MCMC. International Conference on Machine Learning (ICML). 2019.

In equation 4, given fixed P_0 and a long enough MCMC chain, P_T will decorrelate with P_0. How to prevent P_T from collapsing to a delta function? Also intuitively, there should be a weight balancing the two terms of the loss; why a weight of 1 is used?

In equation 8, the function g_{phi} is not continuous because of the indicator function 1(). How do you back-propagate through that function? In the paragraph before Section 3.3, how would you defend the adopted stop-gradient trick?

In the paragraph before Figure 1, how to choose the hyperparameter h? It might not be suitable to set h as the entropy of the prior, as in practice prior and posterior might be different dramatically.


**Experience Assessment:**

I have published one or two papers in this area.

**Review Assessment: Checking Correctness Of Derivations And Theory:**

I carefully checked the derivations and theory.

**Review Assessment: Checking Correctness Of Experiments:**

I assessed the sensibility of the experiments.

**Review Assessment: Thoroughness In Paper Reading:**

I read the paper thoroughly.

---

> ### Author Response · Authors · 2019-11-15
> **Rebuttal Part 1**
>
> Thanks for your valuable feedback. We would like to address concerns and questions in the review as following:
>
> 1) Notation of \pi and \pi^*: The definition of \pi and \pi^* are explicitly stated in the Section 2.1 "MONTE CARLO STATISTICAL INFERENCE". \pi denotes the target distribution and \pi^* is the unnormalised density function of \pi (as mentioned in line 4 in Section 2.1). In the case of Bayesian inference for sampling posterior p(x | y), the target distribution \pi is p(x | y) = p(x, y) / p(y), where p(y) is the normalising constant given observed y, then the unnormalised target density \pi^* is p(x, y) = p(x)p(y | x).
>
> The marginal distribution of the last state of the MCMC chain is denoted by p_T, which is explicitly defined in Equation 3.
>
> 2) Advantages over other combined MCMC and VI:
> *How to optimising MCMC parameters is the main contribution of our ergodic inference (EI) method.*
>
> We compared EI with two relevant hybrid methods of MCMC and VI that *also optimise MCMC parameters*, Hamiltonian variational Inference (HVI)[2] and Hamiltonian Variational Autoencoder (HVAE)[3]. Our experiment results show  the advantage of EI over HVI and HVAE in the both the training efficiency (test log likelihood under the same training time) and the best test log likelihood (See the Table 4 on page 9).
>
> We did not consider other works on MCMC and VI combination, like "A Contrastive Divergence for Combining Variational Inference and MCMC" [1] as the reviewer mentioned, because *the MCMC parameters are not tuned in these methods*. To verify that the MCMC parameters is not optimised in [1], one can simply check on the parameter of variational approximation defined (Equation 3), which does not include any MCMC parameters, and the loss function (Equation 7) in [1]. *For this reason, other works like [1] are less relevant to our EI compared to [2] and [3].*
>
> [1] Francisco J. R. Ruiz and Michalis K. Titsias., 2019.A Contrastive Divergence for Combining Variational Inference and MCMC. International Conference on Machine Learning (ICML).
>
> [2] Salimans, T., Kingma, D.P. and Welling, M., 2015. MCMC and Variational Inference: Bridging the Gap.
>
> [3] Caterini, A.L., Doucet, A. and Sejdinovic, D., 2018. Hamiltonian variational auto-encoder. In Advances in Neural Information Processing Systems (pp. 8167-8177).

---

> > ### Author Response · Authors · 2019-11-15
> > **Part 2**
> >
> > Following Part 1....
> >
> > 3) "...P_T will decorrelate with P_0. How to prevent P_T from collapsing to a delta function? Also intuitively, there should be a weight balancing the two terms of the loss; why a weight of 1 is used?":
> >
> > There are two important details prevent P_T from collapsing to a delta:
> >
> > 1) The convergence of MCMC chain: Well-known in literature like [1] and [2] as mentioned in the paper, the total variation (Theorem 6.53 in [2]) and the KL divergence [1] of P_T to the target distribution \pi decrease *monotonically* after every MCMC transition. Therefore, as long as the ELBO of P_0 is finite, it is guaranteed that the ELBO of P_T is also finite. Therefore, P_T is *impossible* to be a delta distribution (the KL between any non-delta target and a delta distribution is negative infinity), because it is contradictory with the monotonic decay of the KL proved in [2] under the assumption of P_0 with finite ELBO.
> >
> > 2) Maximising the ELBO of P_0 in Ergodic Objective (Equation 4): maximising the ELBO of P_0 prevents P_0 to be a delta. Any P_0 that is a delta distribution has the ELBO of P_0 equal to negative infinity. Therefore, maximising the ELBO of P_0 guarantees that our assumption of P_0 with finite ELBO is true.
> >
> > Combine the two details together, it is not possible under *any circumstances (unless the target is a delta)* for P_T to be a delta distribution.
> >
> > There can be a weight balancing in the loss, but it is most likely to be not useful:
> > 1) with sufficient MCMC steps, the first term has strong dependency with MCMC parameters but very weak dependency with the P_0 parameters
> > 2) In contrast, the second term only depends on the parameters of P_0 but not MCMC parameters at all.
> >
> > Because these two ergodic objective terms depend on two independent sets of parameters separately, the weight balancing strategy does not have much impact in practice. We have verified this by experiments and we are happy to add this in the paper if the paper is accepted.
> >
> > [1] Iain Murray and Ruslan Salakhutdinov. Notes on the KL-divergence between a Markov chain and its
> > equilibrium distribution. preprint, 2008.
> >
> > [2] Christian P. Robert and George Casella. Monte Carlo Statistical Methods (Springer Texts in Statistics).
> > Springer-Verlag New York, Inc., Secaucus, NJ, USA, 2005. ISBN 0387212396.
> >
> > 4) "In equation 8, the function g_{phi} is not continuous because of the indicator function 1()":
> >
> > "Continuous" and "differentiable" are not the same concept. Indicator function is not continuous but differentiable *almost everywhere* (a.e.).
> > Let I(x) be an indicator function in Equation 8, that is I(x) = 1 if p_MH(x) > u otherwise I(x) = 0.
> > Then, we can try to differentiate I(x) w.r.t. x as following three cases:
> > if p_MH(x) > u: I'(x) = \partial_x 1 = 0; (the derivative of constant 1 w.r.t. *any* variable is 0)
> > if p_MH(x) < u: I'(x) = \partial_x 0 = 0; (the derivative of constant 0 w.r.t. *any* variable is 0)
> > if p_MH(x) = u: I'(x) is not defined. (I(x) is not differentiable in this case)
> > Therefore, the derivative of g_{\phi}(x) is well defined a.e. except the case p_MH(x) = u.
> > However, the probability of the case p_MH(x) happens to be *exactly* equal to u is 0, because u is a uniform *continuous* variable between 0 and 1. This explains why g_{\phi} in Equation 8 is differentiable in practice.
> >
> > 5) "how to choose the hyperparameter h?":
> >
> > As motivated in Section 3.1, the hyperparameter h is introduced to avoid pathological cases in optimising MCMC parameter in the case variational approximation P_0 of \pi significantly underestimates the target entropy. This pathology is demonstrated in toy example of correlated Gaussian in Section 3.3. Therefore, the entropy hyperparameter h can be chosen by grid search or any other standard hyperparameter tuning strategy and relatively high value of h should be preferred.
> >
> > As Reviewer 3 mentioned "as the entropy of the prior, as in practice prior and posterior might be different dramatically." This is true. But, priors p(x) are mostly likely to have higher entropy than the posteriors p(x | y) due to the likelihood terms p(y | x). Therefore, the entropy of priors is likely to be a good heuristic choice of h.

---

### Decision · Program_Chairs · 2019-12-19

**Decision:**

Reject

**Comment:**

This paper presents a way of adapting an HMC-based posterior inference algorithm. It's based on two approximations: replacing the entropy of the final state with the entropy of the initial state, and differentiating through the MH acceptance step. Experiments show it is able to sample from some toy distributions and achieves slightly higher log-likelihood on binarized MNIST than competing approaches.

The paper is well-written, and the experiments seem pretty reasonable.

I don't find the motivations for the aforementioned approximations very convincing. It's claimed that encouraging entropy of P_0 has a similar effect to encouraging entropy of P_T, but it seems easy to come up with situations where the algorithm could "cheat" by finding a high-entropy P_0 which leads straight downhill to an atypically high-density region. Similarly, there was some reviewer discussion about whether it's OK to differentiate through the indicator function; while we differentiate through nondifferentiable functions all the time, it makes no sense to differentiate through a discontinuous function. (This is a big part of why adaptive HMC is hard.)

This paper has some promising ideas, but overall the reviewers and I don't think this is quite ready.

---

> ### Author Response · Authors · 2019-12-20
> **Request for Further Clarification on Differentiating Discontinuous Functions**
>
> I appreciate the opinion of the program chair and the reviewers on differentiating discontinuous function, but I have the following questions if would be great if the program chair and reviewers could answer.
>
> First, differentiating through discontinuous MH step is nothing new! It has been used in previous works on gradient based adaptive/auto-tuning MCMC methods, like one of our baseline method, Generalised HMC (GHMC). Generalised HMC was published in the paper "Generalizing Hamiltonian Monte Carlo with Neural Networks" from Daniel Levy, Matthew D. Hoffman, Jascha Sohl-Dickstein, *accepted by ICLR 2018*. GHMC uses gradient of loss function based on HMC samples to tune HMC parameters, which requires differentiation through HMC samples involves discontinuous MH accept-reject step. The question about discontinuity of MH accept-reject in the gradient computation in GHMC appeared as https://openreview.net/forum?id=B1n8LexRZ&noteId=ryeffj94z
>
> Considering existing literature like Levy et al. accepted by ICLR 2018, may I ask what the program chair want to imply by the claim "This (differentiate through a discontinuous function) is a big part of why adaptive HMC is hard."? It is reasonable to assume the program chair of ICLR should be aware of the paper of Levy et al. accepted by ICLR 2018. Then, I cannot help to ask why *our paper is rejected on the ground of no sense to differentiate discontinuous MH accept-reject step*, but the exact same differentiation is used in the GHMC of Levy et al. was accepted by ICLR 2018?
>
> Now let's see the sense of differentiating discontinuous function with random input variables in a formal mathematical point of view.
>
> First, indicator function (the source of discontinuity in the MH step) is equivalent to the heaviside step function, which is differentiable as stated on the wikipedia page https://en.wikipedia.org/wiki/Heaviside_step_function
>
> As mentioned on the wikipedia page, "This (Heaviside step function) was originally developed in operational calculus for the solution of differential equations, ..". So, I can't help to ask why the program chair claim differentiating the indicator function makes no sense, given the fact it is a solution of differential equations in well known literature.
>
> If the words from wikipedia are not convincing/precise enough, I would like to point the program chair to the concept of "almost everywhere". (See the technical explanation from https://en.wikipedia.org/wiki/Almost_everywhere or any textbook on measure theory.)
> You can search "differentiable almost everywhere" for the criteria of functions that is differentiable almost everywhere on the wikipedia page. It is clear that indicator function is differentiable almost everywhere.
>
> The authors of "Generalizing Hamiltonian Monte Carlo with Neural Networks" also mentioned the validity of their differentiation through MH step due to *differentiable almost everywhere* in the discussion of ICLR 2018 see the link https://openreview.net/forum?id=B1n8LexRZ&noteId=ryeffj94z
>
> Finally, let's look into the mathematical nitty-gritty of how *differentiable almost everywhere* gives the validity of differentiating through the discontinuous MH accept-reject step (I have discussed about this in my rebuttal to Reviewer 1, I assume the program chair ignored this):
> Indicator function is not continuous but differentiable *almost everywhere*.
> Let I(x) be an indicator function in Equation 8, that is I(x) = 1 if p_MH(x) > u otherwise I(x) = 0.
> Then, we can try to differentiate I(x) w.r.t. x as following three cases:
> Case 1 p_MH(x) > u: I'(x) = \partial_x 1 = 0; (the derivative of constant 1 w.r.t. *any* variable is 0)
> Case 2 p_MH(x) < u: I'(x) = \partial_x 0 = 0; (the derivative of constant 0 w.r.t. *any* variable is 0)
> Case 3 p_MH(x) = u: I'(x) is not defined. (I(x) is not differentiable in this case)
> Therefore, the derivative of g_{\phi}(x) is well defined a.e. except the case p_MH(x) = u.
> However, the probability of Case 3 where p_MH(x) happens to be *exactly* equal to u is 0, because u is a uniform *continuous* variable between 0 and 1. This explains why g_{\phi} in Equation 8 is differentiable in practice.

---

> ### Author Response · Authors · 2019-12-20
> **Request for the justification on the *seemingly* situations where the algorithm could "cheat" to failure**
>
> I believe the program chair would agree with that it is not fair to reject a paper based on some *seemingly* failure situations. If so, I would like to request the program chair to provide further justification on how to *easily* find those *seemingly* situations our algorithm can fail to converge considering the well-known monotone convergence of MCMC chains.
>
> *In the paper, we have verified empirically it is unlikely to find such situation our algorithm leads straight downhill to an atypically high-density region.*
>
> First, our algorithm is essentially optimising the convergence speed of ergodic Markov chains. It is well known  that ergodic Markov/MCMC chains converge *monotonically* towards the target distribution and the target distribution *only*. This is the most known result in MCMC literature and the foundation of all MCMC methods. Given an initial approximation distribution to the target that is not in the high density area as the optimisation objective constraint Eq. 5, it is simply impossible for the ergodic Markov chain marginal distribution p_T to converge to a delta/nearly delta distribution in the high density area of the target distribution.
>
> Second, hypothetically, it is possible to construct such *seemingly* situations where the algorithm could fail as pointed out the comment above, it is clearly not easy to find such simulations. We have verified our algorithm converges to most common benchmark target distributions, like highly correlated Gaussian in Section 3.3. In our experiment session, the valid convergence of our algorithm is verified on 6 2-d benchmarks without single failure and our algorithm significantly improves the training speed of deep convolutional generative models on MNIST.